# Computational elements based on coupled VO$_2$ oscillators via tunable thermal triggering

Guanmin Li[1], Zhong Wang[1], Yuliang Chen[1], Jae-Chun Jeon[1] ✉ & Stuart S. P. Parkin[1] ✉

Computational technologies based on coupled oscillators are of great interest for energy efficient computing. A key to developing such technologies is the tunable control of the interaction among oscillators which today is accomplished by additional electronic components. Here we show that the synchronization of closely spaced vanadium dioxide (VO$_2$) oscillators can be controlled via a simple thermal triggering element that itself is formed from VO$_2$. The net energy consumed by the oscillators is lower during thermal coupling compared with the situation where they are oscillating independently. As the size of the oscillator shrinks from 6 μm to 200 nm both the energy efficiency and the oscillator frequency increases. Based on such oscillators with active tuning, we demonstrate AND, NAND, and NOR logic gates and various firing patterns that mimic the behavior of spiking neurons. Our findings demonstrate an innovative approach towards computational techniques based on networks of thermally coupled oscillators.

Networks of coupled oscillators provide for highly efficient means of computation—in particular for artificial intelligence tasks such as pattern recognition[1,2]. Two critical aspects for the further development of such systems are mutual interactions and the control of the interactions (coupling strength) between neighboring oscillators for their synchronization. Highly interesting oscillators can be formed from strongly correlated oxide materials that display an insulator-to-metal transition. The controlled oscillation between the low (metal) and high resistance (insulator) states in such materials is possible under an external stimulus, such as current, magnetic field, or electric field[3,4]. Oscillators based on vanadium dioxide (VO$_2$) are of particular interest since the material undergoes an insulating to metallic phase transition near room temperature[5–8]. Coupling between the VO$_2$ oscillators is essential to the operation of the computational network and can be controlled either via external electronic components[9–13] or by thermal links [14].

So far, computational schemes based on the phase relationship between oscillators has been the most common technique used in coupled VO$_2$ networks[10–12]. These schemes rely on binary logic where the two states correspond to the phase of the oscillator (0° or 180°)

relative to a reference oscillator. However, additional electronic elements are required for generating the oscillation and for tuning the coupling among the VO$_2$ devices. This not only increases the complexity of the design of any computational circuit, but also limits the degree of freedom to tune the dynamics of the network during operation. Boolean (logic gates) and non-Boolean (analog) computation techniques have not yet been explored that use oscillation states based on both frequency and amplitude. The latter allows for richer coupling dynamics and, thus, more complex sets of synchronous oscillation states, as introduced in the following sections of this work.

Here we demonstrate that without any additional electronic components, self-sustained VO$_2$ oscillators ranging by more than an order of magnitude in size from 6 μm to 200 nm can be thermally coupled together. We introduce a simple but effective means to actively tune the thermal coupling between VO$_2$ oscillators and show that this can generate a multiplicity of synchronous oscillatory states with distinct frequencies and amplitudes. To achieve this, we insert an additional VO$_2$ nanowire device that

[1]Max Planck Institute of Microstructure Physics, Weinberg 2, 06120 Halle (Saale), Germany. ✉e-mail: jae-chun.jeon@mpi-halle.mpg.de; stuart.parkin@mpi-halle.mpg.de

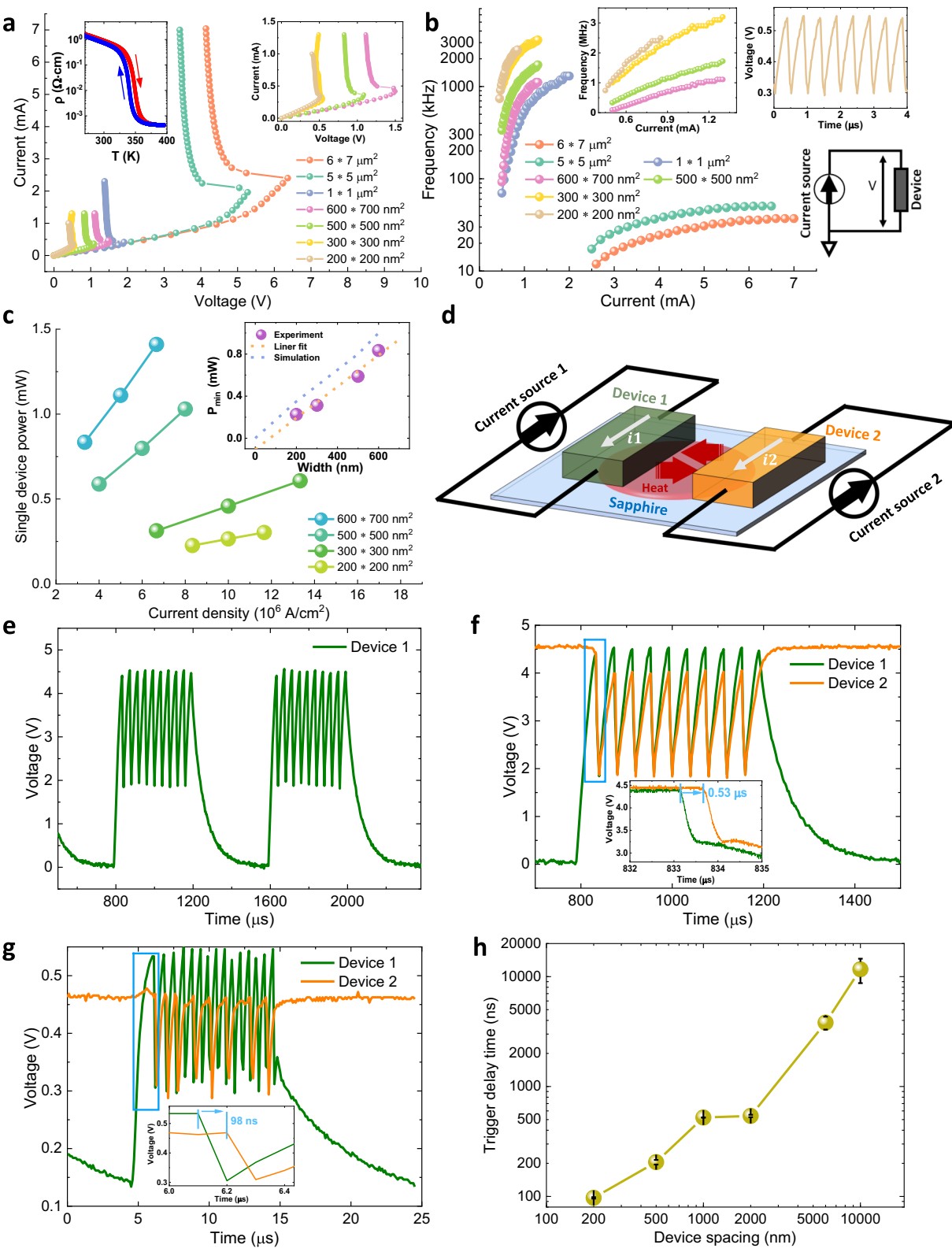

acts as a thermal cell that can actively adjust the synchronous frequency and amplitude of two coupled VO₂ oscillators. The oscillators can be tuned to be in distinct oscillation states via the thermal cell. Two operation modes of the thermal cell, namely V (Voltage) - mode and I (Current) - mode, enable rich and controllable coupling dynamics among oscillators in the system. We utilize various oscillation states generated by pairs of coupled

oscillators to realize 12 basic Boolean logic operations from AND, NAND, and NOR gates using the thermal cell in V-mode. In addition, we find that the thermal cell can induce cascade synchronization among coupled VO₂ cells when operated in I mode, which generates more complex forms of the oscillation states (spiking and bursting patterns) that mimic the behavior of a biological neuron.

**Fig. 1 | Scalability of VO$_2$ oscillators and direct observation of thermal trigger time between two coupled VO$_2$ devices. a** $I$ - $V$ measurements (current is varied while measuring voltage) of single VO$_2$ devices from micron scale (6 × 7 μm$^2$) to nano scale (200 × 200 nm$^2$). $\rho$-$T$ measurement of a VO$_2$ bar (70 × 40 μm$^2$) is shown in the inset left. Inset right shows $I$-$V$ curves for nano scale devices. **b** Oscillation frequency increases with increasing applied current at 295 K. Inset up: Devices at the nano scale. Inset up right: Induced oscillation of a single VO$_2$ device (200 × 200 nm$^2$) for a current of 700 μA. **c,** Single nanoscopic device power versus supply current density. Inset: The minimum power $P_{min}$ to drive the VO$_2$ nano oscillator into stable oscillation for various sizes (Purple circles: experimental data, Orange dashed line: Linear fit of experimental data, Blue dashed line: Simulation results). **d** Schematic illustration of two thermally coupled VO$_2$ devices. **e** Stable oscillation when 400 μs long, 2.8 mA current pulses (separated by 400 μs of zero current) are sent to device 1. **f** Coupled oscillation behavior between device 1 and device 2 (both 5 × 5 μm$^2$) with 2 μm spacing. Device 2 is excited by $I_2$ = 2.3 mA that is lower than the oscillation threshold current (2.5 mA). When device 1 is activated to the oscillation state by a current pulse, device 2 starts to oscillate with the same frequency. Inset (bottom): 0.53 μs to trigger device 2 into oscillation. **g** Coupled oscillation behavior between device 1 and device 2 (both 200 × 200 nm$^2$) with 200 nm spacing. Device 2 is excited by $I_2$ = 460 μA that is lower than the oscillation threshold current (500 μA). When device 1 is activated to its oscillation state by a current pulse (600 μA–20 μs), device 2 can be triggered to oscillate. Inset (bottom): 98 ns to trigger device 2 into its oscillation state. **h** Trigger delay time (error bar shown in black) between two VO$_2$ oscillators for different spacings.

## Results

### Scalability in frequency, power and thermal triggering effect of VO$_2$ oscillators

Figure 1 shows the scalable characteristics of a single and two coupled VO$_2$ oscillators with different sizes. A single device's electrical properties are characterized by $\rho$-$T$ (resistivity versus temperature) and $I$-$V$ (current versus voltage) measurements. The $\rho$-$T$ curve (inset left in Fig. 1a) shows a large change in resistivity at around 340 K that is due to the well-known transition from a monoclinic (M1) insulating phase to a rutile (R) metallic phase (for details of film properties see supplementary Fig. S1 to S3). As shown in Fig. 1a, the VO$_2$ device shows a negative differential resistance (NDR) region, where the voltage drops with increasing current. In this region, the voltage across the device oscillates with a frequency that increases with the magnitude of the applied current, as shown in Fig. 1b. At higher current values the device is heated into the metallic state and no longer oscillates.

The oscillation occurs as follows: When the system is in the high resistance state, applying a d.c. current results in Joule heating ($I^2R$), thereby raising the device temperature and, finally, triggering a phase transition into a low resistance state [15–19]. This lowers the Joule heating and is accompanied by the dissipation of the accumulated heat into the surroundings [20,21]. This leads to cooling and eventually a phase transition back to the high resistance state. The process repeats itself autonomously leading to an oscillatory output voltage (inset top-right in Fig. 1b, the oscillation pattern from a 200 × 200 nm$^2$ device with a frequency $f$ = 2.1 MHz at $I$ = 0.7 mA is shown). With increasing $I$, $f$ also increases. Such a behavior without any external capacitor or resistor only occurs in the $I$-mode and not in the $V$-mode[22]. As this self-sustained VO$_2$ oscillator (driven by constant current) is scaled down to 200 nm in size, a substantial increase in the oscillation frequency (Fig. 1b), as well as a decrease in power (Fig. 1c) are observed. In Fig. 1c inset, we plot the minimum power for driving VO$_2$ cells into a stable oscillation state vs. device size. It clearly shows a linear relationship between power and device size (fit shown by orange dashed line). The linear relationship is also confirmed via simulations (blue dashed line, see supplementary finite element simulation for details).

When two VO$_2$ oscillators are placed close enough together, the heat that is released during one part of the oscillation cycle from one device can trigger the nearby device to oscillate. Interestingly, this mechanism should lead to a time delay (or a phase difference) in the oscillations of the two devices. To explore this phenomenon, pairs of VO$_2$ oscillators with different sizes from 5 × 5 μm$^2$ to 200 × 200 nm$^2$ were fabricated. For a pair of VO$_2$ devices with the size of 5 × 5 μm$^2$ with a 2 μm spacing, device 1 was set in a stable oscillatory state using an above-threshold driving current $I_1$ = 2.8 mA (400 μs pulse), while device 2 was biased with a sub-threshold current $I_2$ = 2.3 mA. As shown in Fig. 1e, device 1 oscillates only within the current pulse window. It is intriguing that device 2 also oscillates at a sub-threshold current, triggered by the thermal energy exchange from the oscillating device 1 (consistent with COMSOL finite element simulations, see supplementary Fig. S4 and S5). Note that the thermal energy exchange can be directly observed as follows: In the first half of the periodic driving

cycle, $I_1$ is large enough to drive device 1 at a stable oscillation (green curve), and device 2 is triggered to oscillate (orange curve) at the same frequency as device 1. Then $I_1$ drops to 0 mA in the second half of the cycle and its voltage, $V_1$, decreases to 0 V, while device 2 recovers to the sub-threshold transition state (high resistance state) and its voltage, $V_2$, increases back to a high level. In the next cycle when $I_1$ is turned on, both devices again start to oscillate at the same frequency, as shown in Fig. 1f. The trigger delay time between the drop of $V_1$ and that of $V_2$ can be seen from the bottom inset. Such a thermal triggering effect can be observed for pairs of devices ranging down to 200 nm in size (Fig. 1g). For the smallest size the trigger delay time is about 98 ns (Fig. 1g bottom inset). The trigger delay time increases with increased spacing, as shown in Fig. 1h (for measurement details see supplementary Fig. S7).

For a single set of VO$_2$ cells (two devices driven by two independent current sources), the further the devices are apart, the weaker is the thermal link (see supplementary Fig. S8). When two devices are far apart, the heat released from one device will dissipate entirely into the substrate before reaching the other device. The thermal coupling strength between the two devices is limited and can only maintain their synchronization within a certain frequency range.

### Realization of VO$_2$ oscillator-based Boolean logic gates by tunable thermal triggering

Now, let us introduce tunable thermal coupling between oscillators by introducing a VO$_2$ thermal cell between two VO$_2$ devices. An optical microscope image and a corresponding circuit diagram are shown in Fig. 2a. The thermal cell is designed to change the ambient thermal environment between device 1 and device 2 (for device set morphology see supplementary Fig. S9). By applying different cell voltages $V_{cell}$ that induce Joule heating, the ambient temperature between device 1 and device 2 can be changed. The higher the cell voltage $V_{cell}$, the higher is the ambient temperature, and, therefore, the lower is the threshold switching voltage (as shown in Fig. 2b inset left) and the higher is the oscillation frequency (for the same current, as demonstrated in Fig. 2b inset right) for a single device. This is because the starting point of the oscillation in the $\rho$ - $T$ hysteresis loop has been biased to a higher temperature closer to $T_c$. Therefore, with an enhanced thermal coupling strength (higher $V_{cell}$), device 1 can be assisted to lock to higher frequencies of device 2 (see discussion in supplementary Fig. S10 to Fig. S17). For the case where the thermal cell is not activated ($V_{cell}$ = 0 V), device 1 was supplied with a constant low current $I_1$ (2.4 mA) and oscillates at a low frequency (11 kHz). By gradually increasing the supply current $I_2$ for device 2, the frequency of device 1 ($f_1$) and the frequency of device 2 ($f_2$) are synchronized until a critical frequency (here 23.5 kHz). When $f_2$ is further increased, due to the limited thermal coupling strength, $f_1$ first drops to a value that is about one-half of $f_2$ and then increases slowly while maintaining a ratio of $f_1$:$f_2$ of roughly 1: 2. In Fig. 2c, the power consumption from device 1 and device 2 when they are oscillating independently, and when they are oscillating with a thermal coupling effect (as shown in Fig. 2d) are plotted, respectively. The thermal energy exchange between device 1

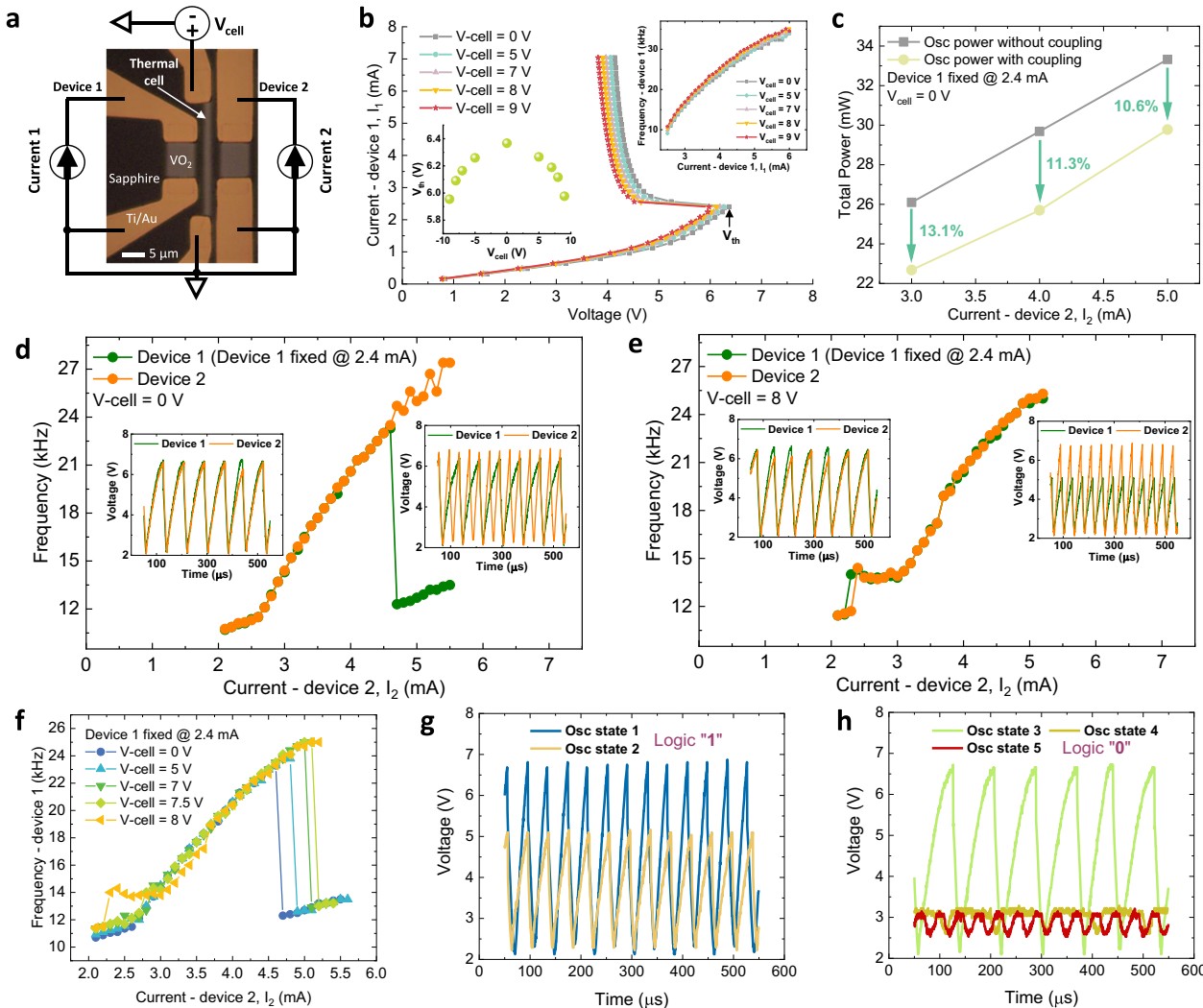

**Fig. 2 | VO₂ oscillators with tunable thermal coupling strength for computation. a** Optical microscopy image of a device set with a schematic illustration of the electrical connections. The device set includes a 1.5 μm wide (21 μm long) VO₂ stripe as a thermal cell between two VO₂ oscillators, each with dimensions of 7 × 6 μm². The separation between device 1 and device 2 is 5 μm. Device 1 and device 2 are driven by two independent current sources $I_1$ and $I_2$, respectively. A voltage source $V_{cell}$ is used to control the thermal cell. **b** $I$ - $V$ measurements (sweep current and measure voltage) of device 1 for different thermal cell voltages ($V_{cell}$). Inset bottom left: The threshold voltage ($V_{th}$) where the device enters the NDR region reduces with increasing $V_{cell}$. Inset top right: $I$ - $f$ measurements (sweep current and measure frequency) from device 1 for different $V_{cell}$. With higher $V_{cell}$, $f$ increases. **c** Comparison of power consumption from device 1 and device 2 under different circumstances at $V_{cell}$ = 0 V. Gray line shows the power summation when they are oscillating independently. Yellow line shows the power summation when they are oscillating with the thermal coupling effect. **d** Frequency locking at $V_{cell}$ = 0 V. $I_1$ is fixed at 2.4 mA while increasing $I_2$. In this case, the frequency locking between device 1 and 2 holds until 23.5 kHz. Inset left: Synchronized waveforms of device 1 ($I_1$ = 2.4 mA) and device 2 ($I_2$ = 2.5 mA). Inset right: Desynchronized waveforms of device 1 ($I_1$ = 2.4 mA) and device 2 ($I_2$ = 5 mA). **e** Frequency locking at $V_{cell}$ = 8 V. The frequency locking fails above 25.3 kHz. Inset left: Synchronized waveforms of device 1 ($I_1$ = 2.4 mA) and device 2 ($I_2$ = 2.5 mA). Inset right: Synchronized waveforms of device 1 ($I_1$ = 2.4 mA) and device 2 ($I_2$ = 5 mA). **f** Comparison of synchronization frequency of device 1 (fixed at 2.4 mA) as $V_{cell}$ is varied (from 0 to 8 V). **g** Oscillation states defined as logic "1". **h** Oscillation states defined as logic "0".

and device 2 helps to reduce the power consumption both in the synchronization region (by 13.1 %) and the non-synchronization region (by 10.6 %), indicating higher energy efficiency (for detailed calculations see supplementary Fig. S18 and S19). For the case where the thermal cell is activated with a very strong thermal coupling strength (Fig. 2e when $V_{cell}$ = 8 V), $f_1$ can now be fully locked to $f_2$ (until both devices are heated to their respective metallic states and don't oscillate anymore. The significant difference between the synchronization behavior of device 1 under weak/moderate ($V_{cell}$ < 8 V) and strong ($V_{cell}$ = 8 V) coupling effects can be observed, as shown in Fig. 2f. Such oscillators with a tunable synchronization behavior can be further scaled down to the nano-scale, as shown in supplementary Fig. S20.

Based on the above tunable thermal coupling mechanism, three different Boolean logic gates (AND, NAND and NOR) are realized by the set of VO₂ devices shown in Fig. 2a. The oscillation state of a VO₂ device, which is represented by the oscillation frequency ($f$) and amplitude ($A$), is taken as state "0" or "1". Here we define the threshold frequency $f_{th}$ as the frequency where device 1 and device 2 desynchronize at $V_{cell}$ = 0 V when $I_1$ is fixed at 2.5 mA while $I_2$ is gradually increasing ($f_{th}$ = 23.5 kHz is taken for the following computation). A frequency lower (higher) than $f_{th}$ is defined as "low (high) frequency". A peak-to-peak value $V_{pk\text{-}pk}$ = 1 V is chosen as the threshold amplitude. An amplitude smaller (larger) than $V_{pk\text{-}pk}$ is defined as "small (large) amplitude". There are 4 kinds of oscillation states; high frequency with large amplitude, high frequency with small amplitude, low frequency

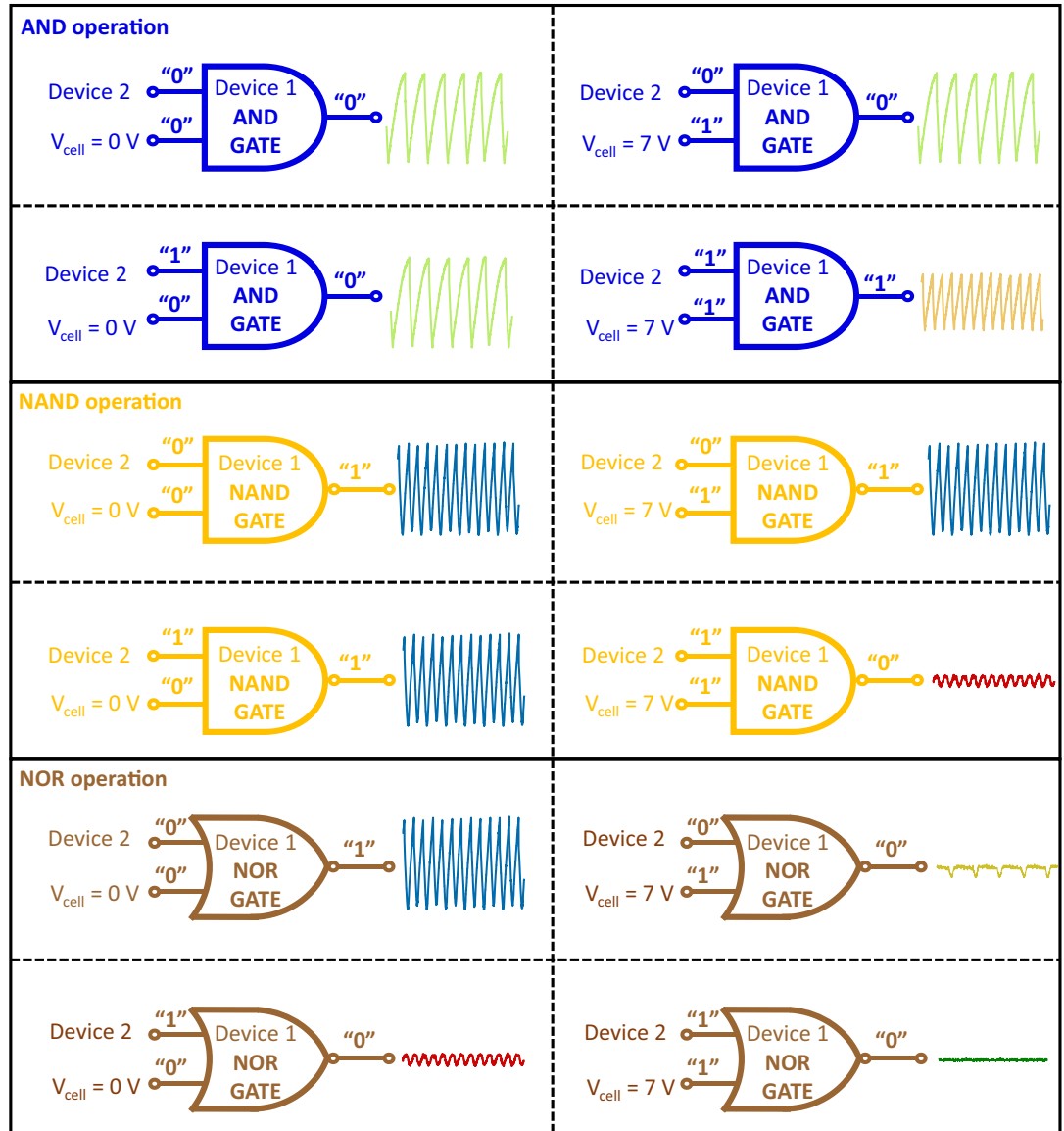

**Fig. 3 | Boolean logic gates based on coupled VO₂ oscillators with tunable thermal coupling strength.** Operations of AND, NAND, and NOR logic gates.

with large amplitude and low frequency with low amplitude. Here, as shown in Fig. 2g and h, only the oscillation state with high frequency and large magnitude is taken as logic "1", while the other states are taken as logic "0" (see supplementary Fig. S21 with detailed discussion of attaining these states).

The oscillation state of device 2 for various $I_2$ is taken as input A, while the thermal cell voltage $V_{cell}$ represents the input B ($V_{cell} = 0(7)$ V stands for B = 0(1)). The current through device 1 ($I_1$) is kept constant during each operation, while the oscillation state of device 1 under input A and input B is taken as the output. 12 Boolean operations from AND gate, NAND gate, and NOR gate are demonstrated in Fig. 3. (Detailed output waveforms from devices 1 and 2 for different logic gates are shown in supplementary Fig. S22. The Boolean-logic calculation table for AND, NAND, and NOR gates is given in supplementary data Table T2).

**Control of firing modes in synchronized cascade VO₂ oscillators**
Operating the thermal cell in the I-mode, in which a constant current ($I_{cell}$) is applied to the thermal cell allows for a very different operational mode of cascade synchronization among the VO₂ oscillators and the thermal cell. This phenomenon occurs because the VO₂ thermal cell will also oscillate for $I_{cell}$ within the NDR region, and the heat periodically released from it strongly links cell 1 and cell 2 to its own oscillation period (see supplementary Fig. S23 to S28 and S6). As shown in Fig. 4a, cell 1, cell 2, and the thermal cell are excited with three independent current sources. With $I_1 = 4$ mA, $I_2 = 3.9$ mA and $I_{cell} = 2.3$ mA, frequency locking among the three devices can be observed in Fig. 4b.

Conceptually, the output waveform of cell 2 is similar to the spiking potential of a neuron when it is stimulated. In biological systems, neurons possess abundant and complex responses to external stimuli so that various spiking neuron models have been established, including tonic spiking and bursting, phase spiking and bursting[23]. As demonstrated in Fig. 4c, the output waveform of cell 2 incorporates four typical regions equivalent to when a neuron transforms from a resting state to an excited state[9,24]. Region-1 corresponds to the resting state when the neuron is not excited and its potential remains at the resting voltage level. Region-2 is when the stimulation arrives and triggers the neuron potential to rise. Region-3 corresponds to the depolarization state where the potential exceeds the threshold and releases a spike. Region-4 is the repolarization and hyperpolarization state (also known as refractory period) where the potential recovers

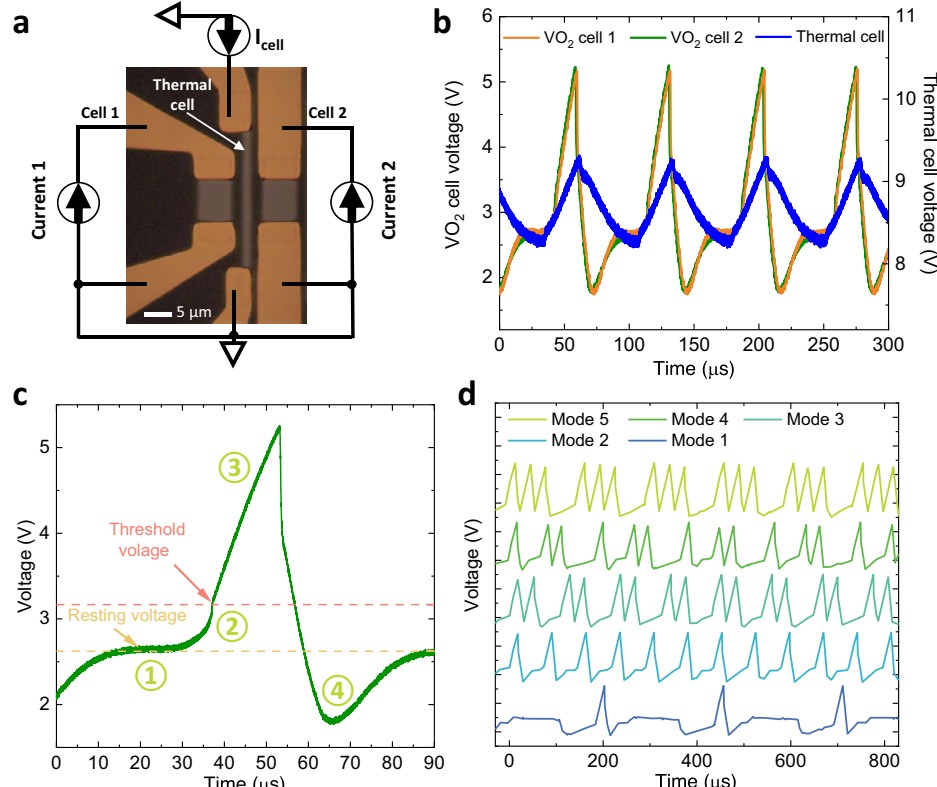

**Fig. 4 | Thermal spike driven VO₂ oscillation with different firing modes.**
**a** Optical image of the VO₂ device set used for the generation of a spiking potential. Cell 1, cell 2, and the thermal cell are excited by three independent current sources. **b** Oscillation waveform when cell 1 is excited at 3.9 mA, cell 2 is excited at 4.0 mA, and thermal cell is activated at 2.3 mA. **c** Output waveform from cell 2 that is generated by the cascade synchronization among three cells. This behavior mimics the generation of the spiking potential in a neuron when it is stimulated; Region-1: Resting state; Region-2: Stimulation arrives; Region-3: Depolarization state; Region-4: Repolarization and hyperpolarization state (also known as refractory period) as discussed in refs. 7,33,34. **d** Different firing modes of the VO₂ neuron by simply changing the current of cell 1 ($I_1$ from 1 mA to 5.2 mA), while keeping the current to cell 2 ($I_2$ = 4 mA) and the thermal cell ($I_{cell}$ = 2.3 mA) fixed.

back to the resting state. By simply adjusting the current through cell 1 (between 1 mA and 5.2 mA) while keeping the currents through cell 2 and the thermal cell constant ($I_2$ = 4 mA, $I_{cell}$ = 2.3 mA), different numbers of spikes (5 different neuron firing modes) can be generated within one firing period from cell 2, as shown in Fig. 4d. Among these 5 firing modes, mode 1 and 2 can be considered as tonic spiking neurons that fire one spike within one period, while mode 3, 4 and 5 can be compared to tonic bursting neurons that fire multiple spikes periodically[25–31].

## Discussion

In this work, we have experimentally demonstrated the scalability of self-sustained VO₂ oscillators driven by a single constant current source. We have also introduced an innovative tunable thermal coupling mechanism between two closely spaced VO₂ oscillators without any external electronic components required. By changing the excitation voltage of the thermal cell placed between two VO₂ oscillators, the thermal coupling strength can be tuned and the range of synchronization frequency of the oscillators can be enlarged. The application of such synchronized oscillators with tunable thermal coupling was demonstrated for Boolean AND, NAND and NOR computations.

This thermally assisted frequency synchronization process can also be considered as being similar to signal propagation in the neural system. The action potential that contains data is modulated in frequency (or amplitude, or both) and propagates from a pre-synaptic neuron to a post-synaptic neuron through a synapse[32–35]. During this process, the action potential is transmitted through the synapse via releasing a neurotransmitter from the axon of the previous neuron to the dendrite of the next neuron [36]. The VO₂ device 1 in Fig. 2a acts as a post-synaptic neuron while device 2 acts as pre-synaptic neuron, respectively. These two VO₂ neurons communicate via an oscillating potential, similar to the above-mentioned neurons in the biological system. The thermal cell is equivalent to the synapse. The post-synaptic neuron (device 1) is supplied with a constant low current and, thus, will only output a low frequency signal when it is not synchronized to the pre-synaptic neuron (device 2). Information is encoded in the domain above the threshold frequency where device 1 and device 2 desynchronize at $V_{cell}$ = 0 V, and is fed into the presynaptic neuron (device 2) as a continuously varying current (frequency-modulated). At $V_{cell}$ = 0 V, there is no synchronization between neurons in the frequency range where information is encoded. As a result, all information from the pre-synaptic neuron (device 2) is lost while propagating to the post-synaptic neuron (device 1). With $V_{cell} \neq 0$ V, meaning that a neural link has been established and two VO₂ neurons are synchronized, the information can be transmitted to the post-synaptic neuron. Such a mechanism can also be applied for frequency modulated data transmission in an analog network for non-Boolean operations.

In conclusion, we have demonstrated VO₂ oscillators that were varied in size from 6 µm to 200 nm. VO₂ oscillators scaled with size over this entire range show substantially reduced energy consumption and higher operating frequency. A pair of thermally coupled VO₂ oscillators are used as Boolean computational logic elements – in which the exchange of thermal energy between oscillators increases the total energy efficiency. Furthermore, we have demonstrated cascade synchronization among three VO₂ cells. By simply changing the current to one of the coupled VO₂ cells, which gives rise to the release of an oscillating thermal energy, five different firing modes including spiking and bursting can be generated from the coupled oscillators.

We believe that such a current-driven firing behavior and a tunable thermal triggering technique can be readily utilized for coding an artificial spiking neural network, in which the output spikes (numbers and firing time) depend on the intensity and spatiotemporal distribution of the input signal[37,38]. Such a network of thermally coupled $VO_2$ oscillators with tunable interactions can also be highly useful for solving different types of computationally hard problems such as pattern classification and combinatorial problems[1,39–41].

## Methods

### Sample and device preparation

$VO_2$ films with thicknesses of about 30 nm (determined by XRR measurements) were deposited on c-plane sapphire [0001] substrates by a PLD (pulsed laser deposition) technique. A KrF excimer laser (Coherent LPX pro) beam with a frequency of 3 Hz was focused onto a $VO_2$ target (99.9 % purity, Plasmaterials) in the PLD chamber under an ambient $O_2$ pressure of 0.020 mbar. The substrate temperature was maintained at 450 °C. Energy and fluence of the laser beam on the target surface were 44 mJ and 587 mJ/cm², respectively. After deposition the sample is cooled down under an ambient $O_2$ pressure of 0.045 mbar. The $VO_2$ thin films were then patterned into devices by conventional optical lithography techniques. A negative tone photoresist (ARN-4340, Allresist), a maskless aligner (Heidelberg MLA 150), and ion beam etching (scia coat 200) were used. For the electron beam lithography, a JEOL EBL apparatus (JBX-8100FS; 100 kV) with a ARN-7520-18 resist was used. Finally, electrodes were prepared by a lift-off method: ~ 77 nm Ti/Au via ion beam deposition (scia coat 200).

### Electrical transport measurement

For temperature-dependent resistance measurements, a physical property measurement system (PPMS) was used with a 4-point resistance measurement scheme and a cooling and warming rate of 5 K/min (from 270 K to 395 K). For the 4-point resistance measurements (Keithley 6221 ac/dc current source and 2182a nanovoltmeter), a 1 μA dc current was applied to the sample to minimize the effect of Joule heating.

For the oscillation state measurements with various types of devices, the measurements were carried out in a multi-probe cryogenic probe station (Lakeshore) with 25 μm diameter W-tips. A current source (Keithely 6221 ac/dc current source) and a source meter (Keithely 2636B) were used to drive the oscillators and the thermal cell. The oscillating voltage was detected by an oscilloscope (DSO5052A, InfiniiVision) with home-built LV codes.

## Data availability

The data that support the findings of this study are available from the corresponding authors upon request.

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

## Acknowledgements

S.S.P.P. acknowledges funding from the European Research Council (ERC) under the European Union's Horizon 2020 research and innovation program (grant agreement No 670166) and the Alexander von Humboldt Foundation in the framework of the Alexander von Humboldt Professorship endowed by the Federal Ministry of Education and Research.

## Author contributions

S.S.P.P. directed the project. G.L., J.J and S.S.P.P. conceived the project and designed the experiments. G.L. and Y.C. prepared the films. G.L. designed and fabricated devices. G.L. characterized the prepared films and devices. G.L. and J.J. conducted electrical transport measurements. J.J. prepared the electrical transport measurement set-ups and programs. Y.C. performed R-T transport measurement. Z.W. performed COMSOL simulations. All authors discussed the results. G.L., J.J., and S.S.P.P. participated in preparing the manuscript.

## Funding

## Competing interests

The authors declare no competing interests.
