## [Peer Review File · Nature Communications]

REVIEWER COMMENTS

Reviewer #1 (Remarks to the Author):

The paper is well prepared , and overall interesting. This is a first demonstration of logical gates with VO₂ oscillators. Even though the results are solid, the paper would get more strength if the authors could elaborate on the possibility to reduce power consumption.

Some extremely low power VO₂ oscillators have been demonstrated, with dimensions at least one order of magnitude smaller. Is the presented work scalable to smaller dimensions and smaller power? Can simulations be done to demonstrate this?

As presented, this paper remains an -interesting- curiosity, and it is hard to believe that any sort of low power computing can be done with circuits presented here.

Comments Fig. 1:

Comments Fig. 2:

Is there any interest to put the thermal cell not exactly in the middle of the structure? Would this affect the synchronisation control of the thermal cell?

Would the thermal cell work without being VO₂?

Fig. 2 Does the comparison take into account the power consumption from the thermal element?

Fig. 3 the authors have selected 5 waveforms for defining 0 and 1. How much is this controlled? Were these waveforms chosen because by doing so the logic gates were working? To which extent this can be engineered?

What if the ambient temperature changes?

Fig. 4 Same questions. To wich extent the control can be designed, and scaled at different frequencies/ device size?

Reviewer #2 (Remarks to the Author):

Authors have fabricated the VO₂ based devices and by exploiting the NDR functionality of these MIT-type devices they have implemented the dynamical oscillator functionality. Moreover, authors have implemented the Boolean logic gates such as NAND, NOR, AND and spiking neuron functionality through the integrated thermal cell between two adjacent VO₂ cells. By changing the mode of operation i.e., I/V-mode of the integrated thermal cell, authors have proposed the input encoding technique for Boolean gates and frequency synchronization of two adjacent VO₂ spiking neurons. Following are some major concerns as a reader point of view:

1. Though fabrication of VO₂ based oscillatory devices is not a new concept and already there are many existing works in literature, also exploiting the thermal coupling effect for locking-mode. Such existing works should be benchmarked and significant advancements should be highlighted in the paper.
2. The NDR functionality can be exploited for oscillations only when there is a series connected load (i.e., load resistance is greater than and equal to a minimum resistance value) is existing between the threshold switching (TS) device and supply voltage so that it can operate the Q-point of the overall circuit in NDR state of the TS device [1]-[2]. However, authors have implemented the oscillator circuit without any series connected load resistance, unlike to existing claims! It infers that the electrode which is supplying a constant current is simultaneously oscillating? Is it so? If not, then how the operating point or Q-point of this circuit is biased in NDR region to generate the sustained oscillations?
3. In introduction section authors have compared physical aspects of MIT (metal-insulator transition) based oscillators, however in-literature; researchers have demonstrated many advance applications using spin-based oscillators – SHNOs/STNOs [1]. Thus authors should provide the challenges of such oscillatory systems? Simultaneously, advantages of VO₂ based oscillators should be highlighted for providing a broader view to a reader?
4. There are majorly phase and frequency based synchronization methods in dynamical systems, however this paper exploits the frequency synchronization method to implement the Boolean logic and dynamical neurons. Here authors should provide an overview – why the one is preferred over the other and what could be the possible challenges?

5. “The latter allows for richer coupling dynamics and thus more complex sets of synchronous oscillation states” – here authors should justify the claim through the evidences from literature?

6. Authors have provided the I-V plots of 7x6 μm^2 size VO₂ device, however the inset in Fig. 1a is for the VO₂ device of size 70x40 μm^2 , so does the larger cross-section based device exhibits the better hysteric in R-T plots? What does it infer of showing the characteristics of two different sizes of devices? Can authors provide the R-T hysteresis of the same device which was used for the I-V plots?

7. In Fig. 1b, what was the input i.e., a DC current of a particular magnitude or a current pulse of fixed width? Though the later section of the paper provides the information about stimulus, however for better understanding of a reader—authors should plot the applied current signal along with the time varying oscillatory output signal for same duration of time-axis?

8. In Fig 1b, Since the Metal-Insulator transition based oscillator circuits are sensitive to ambient temperature along with the input supply [3]. Authors should mention the temperature or extra ambient conditions at which this study was performed?

9. In fig. 1b authors have claimed that the subthreshold current (2.3 mA) also leads to the synchronized oscillations, while the other device is oscillating on the application of above-threshold input current. In line of the same, authors should experimentally demonstrate the range of subthreshold current in device-2, which is producing the sustained and synchronized oscillations?

10. In fig. 2d, authors should explain the method of estimating the power consumption? Was it a source power? Or it was a calculated value? Provide the details, how it was calculated?

11. In fig. 3d, 3e, 3f, authors should provide the output oscillating frequency (i.e., whether its low/high with respecte to the threshold oscillating frequency (f_{th})) and peak-to-peak voltage amplitude (V_{pk-to-Vpk}) for each encoded input case? Since the illustration of only oscillating output i.e., without time and amplitude axis seems confusing to a reader. Please provide all the necessary information.

12. Authors should draw the equivalent schematic diagram of all the proposed Boolean logic gates? Moreover, advantages of these gates over the traditional CMOS based and emerging memristive devices based stateful logic gates should be provided as a benchmarking?

13. What about the fundamental electrical parameters of these gates such as output delay, Fan-In/Fan-Out, Noise margin etc.?

14. Furthermore, the challenges of such Boolean logic gates should also be elaborated?

15. In Fig. 4b, it seems that all three units (two VO₂ devices + thermal cell) are oscillating at the same constant frequency. However, as authors have claimed it as a frequency locking and synchronization state – “...frequency locking among the three devices can be observed in Fig. 4b”. To validate this statement, authors should provide the transient oscillatory output waveforms for all and different current amplitude combinations of I₁, I₂ and Voltage/current amplitude of the thermal cell?

16. The thermal coupling source seems synced-up with the two oscillating devices i.e., device-1, device-2 (Fig. 4b – blue graph) and this is also oscillating with a constant frequency? However, the authors have emulated it as a synaptic unit – “The thermal cell is equivalent to the synapse”. How this is storing and modulating the synaptic-weight?

17. Thermal noise can also be a source of unwanted coupling between two adjacent VO₂ based spiking neurons leading to modulate their spiking rate? How can it be distinguished with the thermal cell coupling? Also throw some light to overcome the thermal noise induced coupling, while implementing such SNN hardware?

18. Industry standard systems are supposed to be working in harsh-ambient conditions. Yuan, Rui, et al. [3] have demonstrated the VO₂ oscillator working even upto 100 °C (i.e., ~373K).

19. However, the proposed VO₂ oscillator works between 295K to 330K temperature range (Extended Fig. 1) after which the proposed oscillatory system may stop working. Consequently, leading to impact the Boolean logic or spiking neuron functionality? Authors should address all of such challenges and suggestions to overcome those challenges should be elaborated?

Minor comments:

1. In fig 1b (inset image) and in supplementary figures, authors have displayed the transient results in time domain from -200 us to 200 us time. Here why the time-axis is varying from -ve range to +ve

range? Generally, it is preferred to illustrate the results either as enlarged view on time axis, else time varying from 0s to a positive value?

2. "Blue line shows the power summation when they are oscillating with the coupling effect: $P_{sum} = \text{Power from device 1 (I1 = 2.4 mA, I2 = 4/5/5.5 404 mA)} + \text{Power from device 2 (I2 = 4/5/5.5 mA, I1 = 2.4 mA)}$ ". Here in both the cases I1 current is constant and I2 current is modulated! Authors should check and verify it.

References:

[1] Yuan, Rui, et al. "A calibratable sensory neuron based on epitaxial VO₂ for spike-based neuromorphic multisensory system." *Nature Communications* 13.1 (2022): 3973.

[2] Shukla, Nikhil, et al. "Synchronized charge oscillations in correlated electron systems." *Scientific reports* 4.1 (2014): 4964.

[3] Torrejon, Jacob, et al. "Neuromorphic computing with nanoscale spintronic oscillators." *Nature* 547.7664 (2017): 428-431.

Reviewer #1 (Remarks to the Author):

The paper is well prepared, and overall interesting. This is a first demonstration of logical gates with VO₂ oscillators. Even though the results are solid, the paper would get more strength if the authors could elaborate on **the possibility to reduce power consumption**.

→ We thank the reviewer for the positive comments on our work and the suggestion to reduce the power consumption of our VO₂ oscillators. We have prepared new devices with much smaller dimensions and have, thereby, successfully demonstrated a nearly 100 x reduction in power by shrinking the size of our devices from 6 microns in our original paper to 200 nm. We include these new results on these new nanoscopic devices in our revised manuscript.

Some extremely low power VO₂ oscillators have been demonstrated, with dimensions at least one order of magnitude smaller. Is the presented work **scalable to smaller dimensions and smaller power**? Can **simulations** be done to demonstrate this?

→ In order to confirm the functionality at smaller dimensions, we have conducted additional experiments on nanoscopic devices. We have fabricated devices, using electron beam lithography techniques, with smaller dimensions ($600 \times 700 \text{ nm}^2$, $500 \times 500 \text{ nm}^2$, $300 \times 300 \text{ nm}^2$, and $200 \times 200 \text{ nm}^2$) that are one order of magnitude smaller than in our original paper. SEM images of typical devices are shown in **Fig. R1**. Furthermore, in order to confirm the validity of the work at the nanoscopic scale, we carried out COMSOL simulations at nanoscopic dimensions (from 600 nm to 50 nm). Note that these simulations agree well with our new experimental data and confirm that the operation power nearly linearly decreases with width of the VO₂ oscillator cell. Both experiment and simulations of the minimum power that drives the VO₂ oscillator into a stable oscillation state for devices with different sizes are shown in **Fig R2**.

Figure R1. SEM images of fabricated nanoscopic devices. **a**, $500 \times 500 \text{ nm}^2$ (spacing = 500 nm), **b**, $300 \times 300 \text{ nm}^2$ (spacing = 300 nm) and **c**, $200 \times 200 \text{ nm}^2$ (spacing = 200 nm). Yellow shadows indicate VO₂ cells (Au contacts in white).

The oscillation frequency increases above 1 MHz with scaling down of the device's width, showing that much faster operation can be achieved by shrinking the size of the device. Moreover, the energy consumption scales down with the device size and

spacing. We have included these additional results from nanoscopic devices in our revised Fig. 1 of the main manuscript (the text is also revised accordingly).

Figure R2. **a**, I-V characteristics of nanoscopic VO₂ devices. Inset (top-right) shows an enlarged view for the devices with the smallest dimensions. (Inset; top-left: ρ vs. T curve of a macroscopic VO₂ bar ($70 \times 40 \mu\text{m}^2$)) **b**, Oscillation frequency increases with increasing applied current within the negative differential resistance region at 295 K. It is noticeable that the oscillation frequency increases with smaller devices (e.g., $\sim 3 \text{ MHz}$ in 300 nm device vs. $\sim 50 \text{ kHz}$ in 5 μm device). Inset shows an enlarged view for the devices with the smallest dimensions. **c**, Single nanoscopic device power versus current density. The inset shows the minimum power P_{min} to drive the VO₂ nano oscillator into a stable oscillation state as a function of device width (Purple circles: experimental data; Orange dashed line: Linear fit of experimental data; Olive dashed line: Simulation

results from COMSOL). **d**, Frequency locking behavior of nanoscopic devices at $I_{cell} = 0$. The device includes a 120 nm wide (2 μm long) VO_2 nano wire as a thermal cell, and two VO_2 oscillators, each with dimensions of $700 \times 600 \text{ nm}^2$. The distance between device 1 and device 2 is 600 nm. The distance between the thermal cell and device 1 (or device 2) is 240 nm. For this measurement, the current to device 1 is fixed at 0.8 mA while the current to device 2 is increased. In this case, the frequency locking between device 1 and 2 holds from 575 kHz to 880 kHz and then breaks down. **e**, Frequency locking behavior at $I_{cell} = 390 \mu\text{A}$. In this case, the frequency locking holds until 1040 kHz. **f**, Tuning of the synchronization frequency range of device 1 as I_{cell} is varied from 0 to 390 μA . Note that the result is similar to Fig. 2d, e and f which shows data for macroscopic devices.

As presented, this paper remains an -interesting- curiosity, and it is hard to believe that any sort of low power computing can be done with circuits presented here.

→ As shown in the additional experiments with nanoscopic devices, we have confirmed that the required power is dramatically reduced with scaling-down of the core device size. In particular, with scaling down to a 600 nm spacing, the operation current density is about $\sim 3 \text{ MA/cm}^2$. Considering recent reports for advanced memory technologies (e.g., SOT-MRAM $\sim 20 \text{ MA/cm}^2$) [R1-1], the operation current density of our device is comparable and promising for compatible circuit development. Furthermore, note that the oscillation frequency increases beyond 3 MHz. This indicates that with a suitable current pulse window, the operation energy can be reduced below 1 nJ. We strongly believe that with proper engineering of materials or with the aid of suitable external circuits, increasing the operation frequency will further lower the operation energy. Also, see our answer to Fig. 4 question below where we compare the power among different computation technologies.

[R1-1] Krizakova, V. *et al.* Spin-orbit torque switching of magnetic tunnel junctions for memory applications. *J. Magn. Magn. Mat.* **562**, 169692 (2022).

Comments Fig. 1:

Comments Fig. 2:

Is there any interest to put the thermal cell not exactly in the middle of the structure? Would this affect the synchronization control of the thermal cell?

→ As shown in the simulations (**Fig. S18**), activating the heat cell thermally perturbs the two coupled VO_2 devices. If the thermal cell is not located exactly in the middle, the device closer to the thermal cell will be activated more strongly than the other one. Such an asymmetric geometry would reduce the synchronization operation window and, therefore, a non-linear response to the input is expected.

During the revision process, we also prepared symmetric and asymmetric device sets. However, due to the different oscillation windows (i.e., we could activate the oscillation from only one device with the thermal cell), we could not successfully synchronize two VO_2 cells that are asymmetrically positioned across the thermal cell.

In a symmetric device set (Fig. R3a – R3c), the thermal cell can be activated ($I_{cell} = 390 \mu\text{A}$) and functions properly so as to synchronize the frequency of device 1 and 2. In an asymmetric device set (Fig. R3d – R3f), however, under the same circumstances (same I_1, I_2, I_{cell}) the thermal cell does not function properly to synchronize the frequency of device 1 and 2.

Figure R3. SEM images of fabricated nanoscopic devices. **a**, The symmetric device set includes a 120 nm wide (2 μm long) VO_2 nano wire as a thermal cell, and two VO_2 oscillators, each with dimensions of $700 \times 600 \text{ nm}^2$. The distance between device 1 and device 2 is 600 nm. The distance between thermal cell and device 1 (or device 2) is 240 nm. Yellow shading indicates the VO_2 cells (Au contacts are shown in white). **b**, Desynchronized waveforms of device 1 ($I_1 = 800 \mu\text{A}$) and device 2 ($I_2 = 1500 \mu\text{A}$) at $I_{cell} = 0 \mu\text{A}$. **c**, Synchronized waveforms of device 1 ($I_1 = 800 \mu\text{A}$) and device 2 ($I_2 = 1500 \mu\text{A}$) at $I_{cell} = 390 \mu\text{A}$. **d**, The asymmetric device set includes a 120 nm wide (2 μm long) VO_2 nano wire as a thermal cell, and two VO_2 oscillators, each with dimensions of $700 \times 600 \text{ nm}^2$. The distance between device 1 and device 2 is 600 nm. The distance between thermal cell and device 1 is 180 nm. The distance between the thermal cell and device 2 is 300 nm. Green shading indicate the VO_2 cells (Au contacts are shown in white). **e**, Desynchronized waveforms of device 1 ($I_1 = 800 \mu\text{A}$) and device 2 ($I_2 = 1500 \mu\text{A}$) at $I_{cell} = 0 \mu\text{A}$. **f**, Desynchronized waveforms of device 1 ($I_1 = 800 \mu\text{A}$) and device 2 ($I_2 = 1500 \mu\text{A}$) at $I_{cell} = 390 \mu\text{A}$.

Would the thermal cell work without being VO_2 ?

→ We attempted to replace the VO_2 cell with a metal wire (Ti/Au). A microscopic picture of the device with a metal thermal cell is shown in Fig. R4. However, such a device combination did not show desirable functions. In particular, since the metallic thermal cell can only heat up the environment, spiking and bursting behavior, shown in Fig. 4, cannot be realized. In fact, using a metallic thermal cell gives the same result as a change in the ambient temperature. We need a device with the essential characteristic of a NDR region in its operation.

Figure R4. Optical microscopy images of VO₂ devices with metallic thermal cells. The thermal cells are made of Ti/Au.

Fig. 2 Does the comparison take into account the power consumption from the thermal element?

→ No, the comparison considers the power when the thermal cell is deactivated, i.e., $V_{cell} = 0$ V. When the thermal cell is activated in the V-mode, extra power consumption from thermal cell can be estimated from the I-V curve of thermal cell as shown in Fig. S7. The total consumption including thermal cell is added to Fig. S19.

Fig. 3 the authors have selected 5 waveforms for defining 0 and 1. How much is this controlled?

Were these waveforms chosen because by doing so the logic gates were working?

→ These waveforms are chosen because the VO₂ device set can generate these wave patterns when the thermal cell is operated in the V-mode. Depending on the input and V_{cell} , 5 distinct waveforms are generated by the device set and we used these waveforms to define computation schemes for AND/NAND/NOR gates.

To which extent this can be engineered?

→ The extent to which Boolean-like logic gates can be controlled and engineered is quite flexible as long as the output frequency and amplitude lie in the range defined as logic “0” or “1”. Let us further elaborate with the ‘AND’ gate: Firstly, let the operation current to device-1 be fixed at 2.5 mA. Then, by changing the current values to device-2 and V_{cell} , the synchronized oscillation state of device-1 can be considered as the output. A similar condition is shown in Fig. 2f (device-1 fixed at 2.4 mA, change V_{cell} and I_2). The calculation/operation rule in our work is defined as follows. For logic “1 (on)”, the oscillation states with high frequency f ($f > f_{th} = 23.5$ kHz) and large magnitude ($V > V_{pk-pk} = 1$ V) are considered. For example, (1) if input B = 1 (on) ($V_{cell} = 7$ V), input A = 0 (off) ($2.0 \text{ mA} \leq I_2 \leq 4.6 \text{ mA}$), output will always be AB = 0 (low f , large V_{pk-pk}); (2) If input B = 1 ($V_{cell} = 7$ V), input A = 1 ($4.6 \text{ mA} \leq I_2 \leq 5.3 \text{ mA}$), output will be AB = 1 (high f , large V_{pk-pk}). (3) If input B = 0 ($V_{cell} = 0$ V), input A = 0 ($2 \text{ mA} \leq I_2 \leq 4.6 \text{ mA}$), output will be AB = 0 (low f , large V_{pk-pk}); finally, (4) If input B = 0 ($V_{cell} = 0$ V), input A = 1 ($4.6 \text{ mA} \leq I_2 \leq 5.3 \text{ mA}$), the output will be AB = 0 (low f , large V_{pk-pk}). Please also

see the answer to Fig. 4.

What if the ambient temperature changes?

→ For VO₂, the oscillation temperature window will be limited by the R-T curve (metal-to-insulator transition temperature) of the device. For example, in our device's case, the operation temperature (T_{OP}) window will be T_{insulator} (~ 280 K) < T_{OP} < T_{metal} (~ 360 K). The operation voltage of the thermal cell will become lower to control the synchronization at higher ambient temperature (but lower than 360 K), and become higher with lower ambient temperature (but higher than 280 K). This is because the thermal cell operated in V-mode is designed to increase the ambient temperature between device 1 and device 2, as explained in detail in the manuscript. Also, note that, at higher ambient temperature, the device will oscillate at higher frequency and smaller amplitude with a given operation current, as explained in **Fig. S8** to **Fig. S14**.

Fig. 4 Same questions. To which extent the control can be designed, and scaled at different frequencies/ device size?

→ The control is quite flexible as mentioned earlier (answer to Fig. 3 above). In fact, we have tried the experiment with various current combinations. When changing the applied current to cell-2 from 4 mA to 3.8 mA, the system can still give the same spiking and bursting behavior (5 different fire modes, as demonstrated in the main text and Fig. R5).

Our work shows that the control of the oscillation state and the link between oscillator cells are flexible but with reproducible output states. Our approach is applicable to reservoir computing for solving complex problems such as combinatorial optimization problems. For instance, Dutta *et al.* [R1-2] have showed that coupled VO₂ oscillators via external circuits can be used as an Ising Hamiltonian solver. In their work, they use the oscillators' phase lock probability via changing the coupling capacitance or resistance to perform the computational task. The overall energy and power for a given solution among different techniques is shown in Table R1-1.

[Table R1-1] [R1-2]

	CPU	GPU	D-Wave	VO ₂ oscillator
Representation of spins	Spins	Spins	Qubits	Oscillator phases
Power	60 W	< 250 W	25 kW	2.56 mW
Energy for solution	14.8 J	< 10 mJ	> 250 MJ	76.8 nJ

However, the system requires external circuit elements to create coupled oscillation states. Our work clearly demonstrates that the thermal link can offer readily applicable coupling strengths between VO₂ cells where the output can be expressed with both frequency and amplitude instead of phase. In principle, such a coupling can be extended to multiple devices which can be used for an Ising Hamiltonian solver [R1-2] or a delay-coupled reservoir computing architecture [R1-3]. Furthermore, spiking neuron firing modes can enrich the dynamic state of the reservoir. In our device, as the delay time depends on the spacing between VO₂ cells, the spatial distribution of VO₂ cells will

play a key role for the functionality.

[R1-2] Dutta et al. An Ising Hamiltonian solver based on coupled stochastic phase-transition nano-oscillators. *Nat. Electron.* **4**, 502 (2021).

[R1-3] Liang, X. et al. Physical reservoir computing with emerging electronics. *Nat. Electron.* *Nat. Electron.* doi.org/10.1038/s41928-024-01133-z (2024).

Figure R5. a, Spiking neuron firing mode-1 at $I_1 = 4.1$ mA (supply current of VO₂ cell 1, yellow line), $I_2 = 3.8$ mA (supply current of VO₂ cell-2, red line) and $I_{cell} = 2.3$ mA (supply current of thermal cell, blue line). **b,** $I_1 = 4.1$ mA, $I_2 = 3.8$ mA and $I_{cell} = 0$ mA. **c,** Spiking neuron firing mode-2 at $I_1 = 3.9$ mA (supply current of VO₂ cell-1), $I_2 = 3.8$ mA (supply current of VO₂ cell-2) and $I_{cell} = 2.3$ mA (supply current of thermal cell). **d,** $I_1 = 3.9$ mA, $I_2 = 3.8$ mA and $I_{cell} = 0$ mA. **e,** Spiking neuron firing mode-3 at $I_1 = 2$ mA

(supply current of VO₂ cell-1), $I_2 = 3.8$ mA (supply current of VO₂ cell-2) and $I_{cell} = 2.3$ mA (supply current of thermal cell). **f**, $I_1 = 2$ mA, $I_2 = 3.8$ mA and $I_{cell} = 0$ mA. **g**, Spiking neuron firing mode-4 at $I_1 = 5$ mA (supply current of VO₂ cell-1), $I_2 = 3.8$ mA (supply current of VO₂ cell-2) and $I_{cell} = 2.3$ mA (supply current of thermal cell). **h**, $I_1 = 5$ mA, $I_2 = 3.8$ mA and $I_{cell} = 0$ mA. **i**, Spiking neuron firing mode-5 at $I_1 = 1$ mA (supply current of VO₂ cell-1), $I_2 = 3.8$ mA (supply current of VO₂ cell-2) and $I_{cell} = 2.3$ mA (supply current of thermal cell). **j**, $I_1 = 1$ mA, $I_2 = 3.8$ mA and $I_{cell} = 0$ mA.

Reviewer #2 (Remarks to the Author):

Authors have fabricated the VO₂ based devices and by exploiting the NDR functionality of these MIT-type devices they have implemented the dynamical oscillator functionality. Moreover, authors have implemented the Boolean logic gates such as NAND, NOR, AND and spiking neuron functionality through the integrated thermal cell between two adjacent VO₂ cells. By changing the mode of operation i.e., I/V-mode of the integrated thermal cell, authors have proposed the input encoding technique for Boolean gates and frequency synchronization of two adjacent VO₂ spiking neurons. Following are some major concerns as a reader point of view:

→ We thank the reviewer for carefully reading our manuscript and pointing out these concerns. Below we answer the concerns raised by the reviewer.

1. Though fabrication of VO₂ based oscillatory devices is not a new concept and already there are many existing works in literature, also exploiting the thermal coupling effect for locking-mode. Such existing works should be benchmarked and **significant advancements** should be highlighted in the paper.

→ As the reviewer points out, there are many works about VO₂ oscillators. For example, VO₂ oscillators are coupled by a resistor [R2-1], a capacitor [R2-2] or by a thermal link [R2-3]. However, we would like to emphasize that these systems face several challenges: (1) In most of these works, the coupling requires external electronic circuit components. In VO₂ systems, researchers have demonstrated tuning the coupling strength via an external capacitor or resistor [R2-1, R2-2]. One of the critical drawbacks is that when adjusting the synaptic connection (coupling strength) of all oscillators during neural network training process, it needs to replace every single capacitor or resistor connected in between, which greatly increases the complexity of the circuit and makes the training process difficult and time consuming. (2) An effective mechanism for real-time tuning of coupling strength between neighboring VO₂ oscillators is missing. Although another study has preliminarily demonstrated thermal coupling effect of two closely located VO₂ oscillators with several micron sized dimensions [R2-3], the coupling strength between them was fixed. When one VO₂ oscillator is operated with a constant current while gradually increasing the current to the other neighbor VO₂ oscillator, the frequency synchronization range of these two VO₂ oscillators is fixed. Also, the applied calculation rule in this previous work [R2-3] is based on the least common multiple of the frequencies of two coupled oscillators, which increases the

complexity of post-processing when extracting the states of the oscillators for computing. (3) Only two coupled states (in-phase or out-of-phase) can be generated and utilized in previously proposed works [R2-1, R2-2]. The challenge of computation based on the phase relationship limits all available coupled states.

Compared to these previous works, our work presents the following significant advantages. (1) Real-time dynamical control of coupling strength is possible without any external electronic components. By simply changing the applied voltage/current to the thermal cell, frequency synchronization range of two coupled oscillators can be actively tuned, as clearly shown in Fig. 1 and Fig. 2. Furthermore, we successfully demonstrated such synchronization in nanoscopic VO₂ oscillators (without using external circuit elements) for the first time. Such an effective tuning mechanism enables a compact and effective network for unconventional computation. (2) Multiple coupled oscillation states can be used for either Boolean-type or analog computational schemes. Compared with coupled states based on phase relationship only [R2-1, R2-2], or frequency ratio only [R2-3], we fully utilize the rich dynamics of the VO₂ oscillator and demonstrate 5 distinct coupled oscillation states based on both frequency and amplitude. Such a multiplicity of synchronous oscillatory states enables 12 different Boolean operations of 3 logic gates, which are realized by the same device-set without changing any electrical layout or circuit elements. Additionally, 5 different firing modes are generated with I-mode operation (thermal cell). This provides a new way to simulate spiking neuron reaction to input stimuli, which can be applied for analog computing in neuromorphic computational schemes.

[R2-1] Corti, E. *et al.* Resistive Coupled VO₂ Oscillators for Image Recognition. *IEEE International Conference on Rebooting Computing (ICRC)*, 1-7 (2018).

[R2-2] Corti, E. *et al.* Time-Delay Encoded Image Recognition in a Network of Resistively Coupled VO₂ on Si Oscillators. *IEEE Electron Device Lett.* **41**, 629-632 (2020).

[R2-3] Velichko, A., Belyaev, M., Putrolaynen, V., Perminov, V. and Pergament, A. Thermal coupling and effect of subharmonic synchronization in a system of two VO₂ based oscillators. *Solid-State Electron.* **141**, 40-49 (2018).

[R2-4] Yi, W. *et al.* Biological plausibility and stochasticity in scalable VO₂ active memristor neurons. *Nat. Commun.* **9**, 4661 (2018).

2. The NDR functionality can be exploited for oscillations only when there is a series connected load (i.e., load resistance is greater than and equal to a minimum resistance value) is existing between the threshold switching (TS) device and supply voltage so that it can operate the Q-point of the overall circuit in NDR state of the TS device [1]-[2]. However, authors have implemented the oscillator circuit without any series connected load resistance, unlike to existing claims! It infers that the electrode which is supplying a **constant current** is simultaneously oscillating?

→ No, this is not the case. Note that the electrode is formed by Au. The oscillation occurs only within the VO₂ cell.

Is it so? If not, then how the operating point or Q-point of this circuit is biased in NDR region to generate the sustained oscillations?

→ The reason why the device supplied with a voltage source V_0 needs an external resistor R_{ext} is that only when the load line of the device ($I = (V_0 - V) / R_{ext}$) intersects with the I-V curve of the device in the NDR region, will an oscillation take place. When a constant current I_0 (I_0 in the NDR region) is applied to the device, the prerequisite for self-sustained oscillation has already been satisfied. We have explicitly explained why the oscillation occurs in our system in the manuscript.

“When the system is in the high resistance state, applying a d.c. current results in Joule heating (I^2R), thereby raising the device temperature and, finally, triggering a phase transition into a low resistance state. This lowers the Joule heating and is accompanied by the dissipation of the accumulated heat into the surroundings. This leads to cooling and eventually a phase transition back to the high resistance state. The process repeats itself autonomously leading to an oscillatory output voltage.”

Note that a similar self-sustained oscillation induced by current source is also reported in [R2-5].

[R2-5] Leroy, J., Crunteanu, A., Givernaud, J., Orlianges, J., Champeaux, C., and Blondy, P. Generation of electrical self-oscillations in two-terminal switching devices based on the insulator-to-metal phase transition of VO₂ thin films. *Int. J. Microw. Wirel. Technol.* 4(1), 101-107 (2011).

3. In introduction section authors have compared physical aspects of MIT (metal-insulator transition) based oscillators, however in-literature; researchers have demonstrated many advance applications using spin-based oscillators – SHNOs/STNOs [1]. Thus authors should provide the challenges of such oscillatory systems? Simultaneously, **advantages of VO₂ based oscillators** should be highlighted for providing a broader view to a reader?

→ Please also see the answer to the 1st question.

Although there have already been a few published works about oscillators realized by STNOs/SHNOs [R2-6 – R2-8], NbO_x [R2-9 – R2-11] and even VO₂ [12 - 14], these systems face several challenges. (1) An effective mechanism for real-time tuning coupling strength between neighboring oscillators is missing. In STNOs/SHNOs system, the tuning of coupling strength (synchronization range) is fixed, which is realized by the spacing between two oscillators [R2-7]. Such a lithographic method is very difficult to apply in a real-time computing system that requires dynamical control of the interactions among oscillators. (2) The control of coupling strength requires external electronic components. In NbO_x [R2-9 – R2-10] and VO₂ [R2-12 – R2-13] systems, researchers have demonstrated the tuning of the coupling strength via external capacitor or resistor. However, when adjusting the synaptic connection (coupling strength) of all oscillators during neural network training, it needs to change every electric component connected in the between, which greatly increases the circuit

complexity and makes the training process difficult and time consuming. (3) Only two coupled states (in-phase or out-of-phase) can be generated and utilized in such systems [R2-9 – R2-13]. The computation based on the phase relationship limits the available coupled states, which only allows binary computational schemes.

The significant advantages of the coupled VO₂ oscillator system proposed in our work over the other oscillation systems are: (1) Real-time dynamical control of coupling strength without any external electronic components are realized. By simply changing the applied voltage/current to the thermal cell, frequency synchronization range of two coupled oscillators can be tuned effectively. (2) I-mode operation to the thermal cell mimics spiking and bursting behavior of biological neuron that requires multiple external resistor/capacitor/inductors to realize in other works [R2-14]. Such an effective tuning mechanism enables a compact network environment for biological system inspired computation tasks. (3) Multiple coupled states can be used for either Boolean-type or analog computational schemes. Compared with coupled states based on phase relationship (phase = 0° or 180° with the same frequency and amplitude), our work demonstrates 5 distinct coupled oscillation states with different frequency and/or amplitude. Such a multiplicity of synchronous oscillatory states enables 12 different Boolean operations of 3 logic gates realized by the same device set without changing any layout nor electrical connection. Besides, 5 different firing modes generated in I-mode operation (thermal cell) provide a new way to simulate spiking neuron reaction to input stimuli.

[R2-6] Chen, T. et al. Spin-Torque and Spin-Hall Nano-Oscillators. *Proc. IEEE* **104**, 1919-1945 (2016).

[R2-7] Mancoff, F., Rizzo, N., Engel, B. et al. Phase-locking in double-point-contact spin-transfer devices. *Nature* **437**, 393–395 (2005).

[R2-8] Romera, M., Talatchian, P., Tsunegi, S. et al. Vowel recognition with four coupled spin-torque nano-oscillators. *Nature* **563**, 230-234 (2018)

[R2-9] Donguk, L. et al. Linear Frequency Modulation of NbO₂-Based Nanoscale Oscillator with Li-Based Electrochemical Random Access Memory for Compact Coupled Oscillatory Neural Network. *Front. Neurosci.* **16**, 939687 (2022).

[R2-10] Lee, D. et al. NbO₂-Based Frequency Storable Coupled Oscillators for Associative Memory Application. *IEEE J. Electron Devices Soc.* **6**, 250-253 (2018).

[R2-11] Rhee, H., Kim, G., Song, H. et al. Probabilistic computing with NbO_x metal-insulator transition-based self-oscillatory pbit. *Nat Commun* **14**, 7199 (2023).

[R2-12] Corti, E. et al. Time-Delay Encoded Image Recognition in a Network of Resistively Coupled VO₂ on Si Oscillators. *IEEE Electron Device Lett.* **41**, 629-632 (2020).

[R2-13] Corti, E. et al. Resistive Coupled VO₂ Oscillators for Image Recognition. *IEEE International Conference on Rebooting Computing (ICRC)*, 1-7 (2018).

[R2-14] Yi, W. et al. Biological plausibility and stochasticity in scalable VO₂ active memristor neurons. *Nat. Commun.* **9**, 4661 (2018).

4. There are majorly phase and frequency based synchronization methods in dynamical

systems, however this paper exploits the frequency synchronization method to implement the Boolean logic and dynamical neurons. Here authors should provide an overview – **why the one is preferred over the other** and what could be the possible challenges?

→ (Also see answers to 1 and 3) Computation based on the phase relationship between oscillators has been the most common technique in previously published works. These schemes rely on binary logic where the two states correspond to the phase of the oscillator (0° or 180°) relative to a reference oscillator. For phase-relation based computation, all oscillators have same frequency. The change of the phase relative to reference oscillator requires changing of the external electronic components (capacitor/resistor), which increase circuit complexity [R2-15]. In this work, on the contrary, the link, i.e., the frequency relative to the reference oscillator (synchronized or not synchronized), between oscillators can be actively tuned via changing the voltage applied on the thermal cell, which greatly simplifies the circuit and enable real-time dynamical control of the synchronization during operation. The possible challenges could be the operation window and thermal noise margin, which we discuss later.

[R2-15] Dutta et al. An Ising Hamiltonian solver based on coupled stochastic phase-transition nano-oscillators. *Nat. Electron.* **4**, 502 (2021).

5. "The latter allows for richer coupling dynamics and thus more complex sets of synchronous oscillation states" – here authors should justify the claim through the evidences from literature?

→ By changing the operation from V-mode to I-mode, coupled VO₂ oscillators generate distinct signal pulses. When a VO₂ thermal set is set to oscillation state via a d.c. current, it releases periodic thermal effects thus creating a bursting thermal link between VO₂ oscillators. This creates more complex sets of synchronous oscillations states and the generated signal from such a bursting thermal link mimics the spiking behavior of a biological neuron. As we utilize both frequency and amplitude of the coupled oscillation state, compared with computation based on phase relationship, it allows richer coupling dynamics between oscillators. Thus far, to our knowledge, there are hardly any published works that have shown the range of the frequency synchronization of two oscillators and that their oscillation amplitude can be tuned without any extra electronic components. Our work, for the first time, shows that two coupled oscillators can be used for carrying out logic gate operations by actively tuning both synchronization frequency and amplitude. Such rich dynamics can be used for both Boolean and non-Boolean (analog) computation techniques.

We have modified the sentence in the main text as follows.

“The latter allows for richer coupling dynamics and thus more complex sets of synchronous oscillation states as introduced in the following sections of this paper.”

6. Authors have provided the I-V plots of 7x6 μm^2 size VO₂ device, however the inset in Fig. 1a is for the VO₂ device of size 70x40 μm^2 , so does the larger cross-section based device exhibits the better hysteric in R-T plots? What does it infer of showing the

characteristics of two different sizes of devices? Can authors provide the **R-T hysteresis of the same device** which was used for the I-V plots?

→ We have measured R-T curves from devices with dimensions of $7 \times 6 \mu\text{m}^2$, $1 \times 1 \mu\text{m}^2$, and $600 \times 700 \text{nm}^2$, as shown in **Figure R6**. These additional results are added to **Fig. S20**. It shows that the R-T hysteresis loop becomes smaller with decreasing dimensions due to a spatial confinement effect [R2-16]. Note that the resistivity is lower in the insulating state (smaller R-T hysteresis loop) which reflects the smaller oscillation amplitude and higher oscillation frequency shown in **Fig. 1a** and **b**.

[R2-16] Hattori, A. N. et al. Investigation of Statistical Metal-Insulator Transition Properties of Electronic Domains in Spatially Confined VO₂ Nanostructure. *Crystals*. **10(8)**, 631 (2020).

Figure R5. R-T curves of VO₂ devices with different sizes.

7. In **Fig. 1b**, what was the input i.e., a DC current of a particular magnitude or a current pulse of fixed width? Though the later section of the paper provides the information about stimulus, however for better understanding of a reader—authors should plot the applied current signal along with the time varying oscillatory output signal for same duration of time-axis?

→ In **Fig. 1b** the input is a d.c. current of a particular magnitude (3 mA). The figure caption has been modified to make it clearer.

8. In **Fig 1b**, Since the Metal-Insulator transition based oscillator circuits are sensitive to ambient temperature along with the input supply [3]. Authors should mention the **temperature or extra ambient conditions** at which this study was performed?

→ The measurement is taken at 295K. This has been added to the graph for clarification.

9. In **fig. 1b** authors have claimed that the subthreshold current (2.3 mA) also leads to

the synchronized oscillations, while the other device is oscillating on the application of above-threshold input current. In line of the same, authors should experimentally demonstrate **the range of subthreshold current in device-2**, which is producing the sustained and synchronized oscillations?

→ The subthreshold current range is from 2.3 mA to 2.5 mA, as shown in **Fig. 1**. Above this range device 2 will start to oscillate by itself (**Fig. 1b**). Below this range device 2 can't be triggered by device 1.

10. In **fig. 2d**, authors should explain the method of estimating the power consumption? Was it a source power? Or it was a calculated value? Provide the details, how it was calculated?

→ The power is calculated as follows: The oscillating voltage waveform under a certain current value is collected. Then the voltage value is averaged over time (500 μ s) then multiplied by the current value to obtain the average power consumption. This clarification has been added to **Fig. S19**

11. In **fig. 3d, 3e, 3f**, authors should provide the **output oscillating frequency** (i.e., whether its low/high with respect to the threshold oscillating frequency (fth)) and **peak-to-peak voltage amplitude** ($V_{pk-to-Vpk}$) for each encoded input case? Since the illustration of only oscillating output i.e., without time and amplitude axis seems confusing to a reader. Please provide all the necessary information.

→ Yes, the data is available and has been added to **Extended Data Table. 1**.

12. Authors should draw the equivalent schematic diagram of all the proposed Boolean logic gates? Moreover, **advantages of these gates over the traditional CMOS based** and emerging memristive devices based stateful logic gates should be provided as a benchmarking?

→ The equivalent schematic diagram of all the proposed Boolean logic gates is illustrated in **Fig. 3d to f**.

The advantages of the proposed Boolean logic gates are: 1. The Circuit complexity can be reduced. Because traditional CMOS based AND/NAND/NOR gates have different combinations of NMOS and PMOS. In order to implement different logic functions, their circuits have to be designed separately and the transistors will be wired differently. Operating different logic functions will have to be carried out in different parts of the chip. However, VO_2 based logic gates (AND/NAND/NOR) proposed in Fig. 3 all share the same layout (two devices coupled with a thermal cell), which means that different logic gate operations can be carried out exactly at the same device set without changing any layout design or electrical connection; 2. Multi analog bits transmission is enabled. In traditional CMOS based logic gates only bit "0" and bit "1" can be input or output, which is binary and well-known as digital system. In VO_2 based logic gates, however, the input and output can be analog depending on the oscillating frequency. Take the VO_2 based AND gate in Fig. 3d as an example: When device 1 and device 2 are synchronized ($V_{cell} = 7$ V), the output frequency of device 1 will always be the same as device 2. When the input information is modulated in different frequency (digital

scheme: only low f and high f ; analog scheme: continuous change of f), the gate will always be able to transmit different FM (frequency-modulated) signals, which covers a certain bandwidth (synchronization range). Such analog bit transmission system, which can be very hard to implemented by traditional CMOS based logic gates, provides a new way to develop computational system such as artificial neural network where the input and output are not binary anymore. Most of the works reporting logic units realized by memristive devices compute either in binary (high/low resistance state) or with phase-relationship (0° or 180° relative to a reference oscillator, still binary). Also, previous work has shown the advantage of lower power consumption of coupled-VO₂-based Ising machine over traditional CMOS-based CPU when solving Combinatorial optimization problems (non-deterministic polynomial time (NP)-hard complexity class) [R2-15] – also see table (R1-1) for comparison between different computation techniques.

[R2-15 = R1-2] Dutta et al. An Ising Hamiltonian solver based on coupled stochastic phase-transition nano-oscillators. Nat. Electron. **4**, 502 (2021).

13. What about the fundamental electrical parameters of these gates such as output delay, Fan-In/Fan-Out, Noise margin etc.?

→ The output delay is considered to be the trigger time delay demonstrated in **Fig. 1h**, which lies in the range from about 98 ns (devices are separated by 200 nm) to 4 μ s (devices are separated by 6 μ m). From our experiments (Fig. 1 and Fig. 2), our devices exhibit a direct output amplitude > 0.5 V (without any external amplifiers). With further scaling-down, the output could be reduced to ~ 0.1 V which indicates ~ 20 mV of noise margin with 20% – 80% V_{pk-pk} condition. Regarding Fan-In/Fan-Out, although it is not our major focus of this work, it is definitely an interesting aspect to discuss for technological applications in the future. Unlike conventional digital memory technologies, in our oscillator system, d.c. input signal (current) becomes a.c. output signal (frequency, amplitude). Therefore, conventional Fan-In/Out concept is not directly applicable to such analog system.

14. Furthermore, the challenges of such Boolean logic gates should also be elaborated?

→ Challenges of such Boolean logic gates could be: Ambient thermal fluctuations that could cause oscillation frequency and amplitude to change, as mentioned in the SI. Such problems can be addressed by using good thermal insulation materials to seal the devices.

15. In Fig. 4b, it seems that all three units (two VO₂ devices + thermal cell) are oscillating at the same constant frequency. However, as authors have claimed it as a frequency locking and synchronization state – "...frequency locking among the three devices can be observed in Fig. 4b". To validate this statement, authors should provide the **transient oscillatory output waveforms** for all and different current amplitude combinations of I₁, I₂ and Voltage/current amplitude of the thermal cell?

→ The transient oscillatory output waveforms for all and different current amplitude

combinations of I_1 , I_2 and voltage of the thermal cell (V-mode) for logic gates operations are shown in **Extended Data Fig. 4**. The transient oscillatory output waveforms for all and different current amplitude combinations of I_1 , I_2 and current of the thermal cell (I-mode) for mimicking spiking and bursting behavior of biological neuron are shown in **Extended Data Fig. 5** to **Extended Data Fig. 9**.

16. The thermal coupling source seems synced-up with the two oscillating devices i.e., device-1, device-2 (Fig. 4b – blue graph) and this is also oscillating with a constant frequency? However, the authors have emulated it as a synaptic unit – **"The thermal cell is equivalent to the synapse"**. **How this is storing and modulating the synaptic-weight?**

→ In the text, the thermal cell is referred to as a synapse when it is operated in V-mode. As stated in the text

“The VO₂ device 1 in Fig. 3a acts as a post-synaptic neuron while device 2 acts as pre-synaptic neuron, respectively. These two VO₂ neurons communicate via an oscillating potential, similar to the above-mentioned neurons in the biological system. The thermal cell is equivalent to the synapse... With $V_{cell} \neq 0$ V, meaning that a neural link has been established and two VO₂ neurons are synchronized”.

The connection strength between neurons (i.e., synaptic weight) is represented as the V_{cell} value, which is constant during the operation. The oscillating thermal cell operated in I-mode shown in Fig. 4b is only used to generate the spiking/bursting behavior of VO₂ cell 2.

17. **Thermal noise** can also be a source of unwanted coupling between two adjacent VO₂ based spiking neurons leading to modulate their spiking rate? How can it be distinguished with the thermal cell coupling?

→ We thank the reviewer for pointing this out. Since our device concept is based on thermal coupling, thermal noise can be critical for its functionality. When there is no thermal noise or subtle thermal noise, constant output frequency and amplitude from certain input will be expected, as long as the noise induced V fluctuation is within the noise margin discussed in Q13. However, when the system is influenced by unwanted thermal noise, i.e., large ambient temperature fluctuation, the output frequency and amplitude will be changed as demonstrated in the SI for different temperatures. We could consider two different scenarios. (1) When the thermal noise is persistent, i.e., permanent change in ambient temperature, the spiking rate will be permanently changed at given input current. In such case, one could find optimum range by adjusting input current. (2) In case of temporal thermal noise, i.e., thermal spike, correct rate/solution can be found by recurrent mapping since the signal will be sampled over a certain period of time so the impact from noise will be averaged out and the final output will maintain good signal-to-noise ratio.

Also throw some light to overcome the thermal noise induced coupling, while implementing such SNN hardware?

→ For hardware applications, good thermal isolation of the devices is desirable so that

it is tolerable to ambient temperature fluctuations. The time period used to sample the signal for calculating spiking rate can be properly chosen to reach a good compromise between signal process speed and signal-to-noise ratio. Firing rate fluctuating in a small range can be tolerated or rectified by the algorithm to increase system robustness. Also, see the answer to Q17.

18. Industry standard systems are supposed to be working in harsh-ambient conditions. Yuan, Rui, et al. [3] have demonstrated the VO₂ oscillator working even up to 100 °C (i.e., ~373K).

→ This is not correct. In Yuan, Rui et al's paper, the original text is "As the temperature increases, the resistance of the temperature sensor decreases, leading to increased spiking frequency". Notice that the VO₂ oscillator device in their work is at ambient temperature. The temperature range (from 30 to 100°C) in Fig. 4h (Yuan, Rui's paper) corresponds to the temperature of a sensor not the VO₂ device itself. The sensor is a resistor that is connected in series with the VO₂ device. The change in the resistance of this sensor, an external resistor (R_{scaling}), which is connected to VO₂ device, tunes the frequency of the VO₂ oscillator (Fig. 4g). This does not mean that the VO₂ oscillator can work at 100°C. In fact, VO₂ has a phase transition temperature at around 340 K (67°C). Above this temperature, VO₂ will become fully metallic where no oscillations are possible.

19. However, the proposed VO₂ oscillator works between 295K to 330K temperature range (Extended Fig. 1) after which the proposed oscillatory system may stop working. Consequently, leading to impact the Boolean logic or spiking neuron functionality? Authors should address all of such challenges and suggestions to overcome those challenges should be elaborated?

→ (Also see Q17) The most important requirement for using phase transition materials (NbO_x, VO₂...) as an oscillator is to control the working temperature to be below their critical phase transition temperature. When the system is influenced by thermal perturbations, i.e., ambient temperature fluctuations, a change in frequency and amplitude from the output can be observed (as demonstrated in the SI for various temperatures). In order to overcome the challenges mentioned above, good thermal insulation materials can be used for packaging the devices to protect them from overheating by external high temperatures and also, they wouldn't then be affected by ambient temperature fluctuations.

Minor comments:

1. In fig 1b (inset image) and in supplementary figures, authors have displayed the transient results in time domain from -200 us to 200 us time. Here why the time-axis is varying from -ve range to +ve range? Generally, it is preferred to illustrate the results either as enlarged view on time axis, else time varying from 0s to a positive value?

→ The time-axis varies from - μs to + μs range is because the data is directly exported from the oscilloscope (the time axis in oscilloscope is in - μs to + μs range). The pictures have been modified to have the time axis starting from 0.

2. "Blue line shows the power summation when they are oscillating with the coupling effect: $P_{sum} = \text{Power from device 1 (I}_1 = 2.4 \text{ mA, I}_2 = 4/5/5.5 \text{ mA}) + \text{Power from device 2 (I}_2 = 4/5/5.5 \text{ mA, I}_1 = 2.4 \text{ mA})$ ". Here in both the cases I_1 current is constant and I_2 current is modulated! Authors should check and verify it.

→ Yes, because when there is a coupling effect, it means both device 1 and device 2 are oscillating, so that their oscillation states can be influenced by the heat periodically released from the neighboring device, as explained in the paper. So, when probing power consumption with the coupling effect both devices have to be turned on. Here device 1 is fixed at 2.4 mA, and the total power consumption of both devices when the current from device 2 is gradually increased from 4 to 5.5 mA is calculated. That's why I_1 current is constant and I_2 current is modulated.

References:

- [1] Yuan, Rui, et al. "A calibratable sensory neuron based on epitaxial VO₂ for spike-based neuromorphic multisensory system." *Nature Communications* 13.1 (2022): 3973.
- [2] Shukla, Nikhil, et al. "Synchronized charge oscillations in correlated electron systems." *Scientific reports* 4.1 (2014): 4964.
- [3] Torrejon, Jacob, et al. "Neuromorphic computing with nanoscale spintronic oscillators." *Nature* 547.7664 (2017): 428-431.

REVIEWERS' COMMENTS

Reviewer #1 (Remarks to the Author):

My main comments on the first version of this manuscript were about the scalability of the proposed coupled oscillators. The authors included experimental data for sub micron scale of VO₂ oscillators. The demonstration of coupled oscillations in small devices is really a significant advance.

The response to reviewers is fine for me, and answered my questions.

The last suggestion would be to give an approximate size for a transistor + RC oscillator in a CMOS process compared to the current VO₂ oscillators.

The useable temperature window is still vague in my understanding of the paper.

Reviewer #2 (Remarks to the Author):

Authors have revised the manuscript by addressing the previous queries one-by-one. The current and revised form of the manuscript has been significantly improved from the previous version. I recommend to consider the manuscript for publication in its current form.

Reviewer #1 (Remarks to the Author):

My main comments on the first version of this manuscript were about the scalability of the proposed coupled oscillators. The authors included experimental data for sub micron scale of VO₂ oscillators. The demonstration of coupled oscillations in small devices is really a significant advance.

The response to reviewers is fine for me, and answered my questions.

→ **We thank the reviewer for accepting our revised manuscript.**

The last suggestion would be to give an approximate size for a transistor + RC oscillator in a CMOS process compared to the current VO₂ oscillators.

→ **We have looked at prior studies to compare the approximate size of a transistor + RC oscillator in a CMOS process [R1, R2]. These show that typical sizes are 350 nm CMOS + 1 um x 1.5 um (PDIFF) [R1] and 78 um x 132 um [R2]. Since thermally coupled VO₂ devices do not require external circuit elements, current VO₂ oscillators will be ~ one order smaller size compared to CMOS based devices. However, since such technological details are not the main focus of our work, we did not include this in our manuscript.**

[R1] F. R. G. Cruz, M. A. E. Latina and Wen-Yaw Chung, "CMOS RC oscillator using 0.35 micron for portable mosquito-repel circuit," TENCON 2015 - 2015 IEEE Region 10 Conference, Macao, China,

[R2] B. Wang, M. -L. Ko and Q. Yan, "A high-accuracy CMOS on-chip RC oscillator," 2010 10th IEEE International Conference on Solid-State and Integrated Circuit Technology, Shanghai, China, 2010

The useable temperature window is still vague in my understanding of the paper.

→ **The usable temperature window of our device is near to the insulator-to-metal phase transition temperature i.e. 290 K ~ 350 K in our device's case (see $\rho - T$ curve in Fig. 1a).**

Reviewer #2 (Remarks to the Author):

Authors have revised the manuscript by addressing the previous queries one-by-one. The current and revised form of the manuscript has been significantly improved from the previous version. I recommend to consider the manuscript for publication in its current form.

→ **We thank the reviewer for accepting our revised manuscript.**